# The Link between Purine Metabolism and Production of Antibiotics in *Streptomyces*

**DOI:** 10.3390/antibiotics8020076

**Published:** 2019-06-06

**Authors:** Smitha Sivapragasam, Anne Grove

**Affiliations:** Department of Biological Sciences, Louisiana State University, Baton Rouge, LA 70803, USA; smithatop100@gmail.com

**Keywords:** c-di-GMP, guanosine and (p)ppGpp, purine salvage, secondary metabolism, Streptomycetes, stringent response

## Abstract

Stress and starvation causes bacterial cells to activate the stringent response. This results in down-regulation of energy-requiring processes related to growth, as well as an upregulation of genes associated with survival and stress responses. Guanosine tetra- and pentaphosphates (collectively referred to as (p)ppGpp) are critical for this process. In Gram-positive bacteria, a main function of (p)ppGpp is to limit cellular levels of GTP, one consequence of which is reduced transcription of genes that require GTP as the initiating nucleotide, such as rRNA genes. In Streptomycetes, the stringent response is also linked to complex morphological differentiation and to production of secondary metabolites, including antibiotics. These processes are also influenced by the second messenger c-di-GMP. Since GTP is a substrate for both (p)ppGpp and c-di-GMP, a finely tuned regulation of cellular GTP levels is required to ensure adequate synthesis of these guanosine derivatives. Here, we discuss mechanisms that operate to control guanosine metabolism and how they impinge on the production of antibiotics in *Streptomyces* species.

## 1. Introduction

Bacteria experience constant challenges, either in the environment or when infecting a host. They utilize various mechanisms to survive such stresses, which may include changes in temperature, pH, or oxygen content as well as limited access to carbon or nitrogen sources. A stress response that is mobilized to deal with events such as nutrient limitation is the stringent response. The purpose of this response is to focus cellular resources on survival, as opposed to growth [1]. To achieve this outcome, bacteria synthesize the hyperphosphorylated nucleotides guanosine pentaphosphate and guanosine tetraphosphate, collectively called (p)ppGpp, from GTP/GDP and ATP. Based on the critical role of (p)ppGpp, it is also called an alarmone, as it responds to starvation or stress and conveys such distressing circumstances to the cellular machineries in order to conserve resources. Exposure to challenging environmental conditions may also induce formation of biofilms, protective bacterial communities that confer a number of fitness advantages, including resistance to nutrient deprivation or antimicrobial agents. The second messenger cyclic-di-GMP (c-di-GMP), which is synthesized from two molecules of GTP, is intimately involved in the switch between a motile planktonic lifestyle and a sessile biofilm community in unicellular bacteria [2]. In multicellular Actinomycetes, such as *Streptomyces* spp., c-di-GMP is involved in controlling developmental processes [3]. Collectively, these signaling molecules require the presence of purines to sustain their synthesis under conditions of stress and starvation, circumstances under which the bacterial cell must conserve essential nutrients. Purine salvage pathways are, therefore, expected to be important for the generation of substrates for (p)ppGpp and c-di-GMP synthesis. Additionally, members of the genus *Streptomyces* respond to changes in environmental conditions by a morphological differentiation and multicellular development that may be closely tied to their production of secondary metabolites, including a wide range of antibiotics. Indeed, two thirds of clinically relevant antibiotics such as chloramphenicol, streptomycin, and tetracycline are synthesized by *Streptomyces* spp. [4]. In this review, we discuss how purine metabolism affects antibiotic production, with an emphasis on the stringent response.

## 2. Stringent Response and Morphological Differentiation

Nutrient limitation or starvation induces the stringent response in the vast majority of bacteria. This response was first described in *Escherichia coli* and shown to involve reduced accumulation of stable RNAs (rRNA and tRNA); this stringent response was ‘relaxed’ by mutation in the gene accordingly named *relA*. Stringent response depends on a transient increase in the level of (p)ppGpp, also known as the magic spot [5]. Members of the RSH (RelA/SpoT Homolog) protein superfamily are responsible for (p)ppGpp synthesis from ATP and either GTP or GDP and for the degradation of (p)ppGpp to GTP or GDP and pyrophosphate, with some members possessing both enzymatic functions. The activity of different RSH family proteins is controlled by distinct mechanisms, including interaction with idling ribosomes in many cases, allowing responses to various stresses [1]. The stringent response may be initiated by limitation of nutrients such as amino acids, nitrogen, phosphorous, and carbon sources. Once (p)ppGpp accumulates, it exerts global changes designed to downregulate transcription of genes linked to growth (e.g., ribosome biogenesis) and upregulation of genes required for survival (e.g., nutrient acquisition and stress responses), and it also directly binds a number of proteins to regulate their activity [6]. While specific binding targets may differ between Gram-negative and Gram-positive species, (p)ppGpp has been shown to regulate replication, transcription, translation and GTP biosynthesis by binding to proteins that participate in these processes, including DNA primase, RNA polymerase, small GTPases, enzymes involved in purine biosynthesis, and transcriptional regulators [7,8,9,10,11].

Considerable differences exist between bacterial species with regard to both the mode of action and metabolism of (p)ppGpp. In Gram-negative species such as *E. coli*, ribosomes sense the uncharged tRNAs at the ribosomal A site during amino acid starvation, causing protein synthesis to stall and resulting in the activation of RelA [12]. This leads to synthesis of (p)ppGpp, which acts as an allosteric regulator of RNA polymerase. The (p)ppGpp binds together with the regulatory protein DksA; this leads to inhibition of transcription from stable RNA promoters and promoters driving expression of genes encoding, for example, ribosomal proteins, and it mediates an activation of genes critical for cell survival [13,14]. This mechanism is likely to be conserved among other proteobacteria [14].

The stringent response in Gram-positive bacteria is not mediated by direct binding of (p)ppGpp to RNA polymerase [15,16]. In *Bacillus subtilis*, (p)ppGpp regulates intracellular GTP levels, even during normal growth, and dysregulated GTP homeostasis leads to cell death [17,18]. Two mechanisms contribute to the observed reduction in GTP levels during the stringent response, namely, consumption of GTP for the purpose of (p)ppGpp synthesis and inhibition of GTP biosynthesis. GTP can be synthesized by both expensive *de novo* and less resource-demanding salvage pathways, and (p)ppGpp binds directly to several proteins that participate in these pathways to inhibit their activity [10,18]. Figure 1 summarizes purine metabolism, identifying key regulated enzymes participating in synthesis of GTP; *de novo* synthesis initiated by GuaB-catalyzed conversion of inosine monophosphate (IMP) to XMP, or formation of XMP by purine salvage, which requires xanthine dehydrogenase (Xdh). As a consequence of reduced cellular GTP levels, transcription from genes whose initiating nucleotide is GTP is decreased, thereby accounting for the reduced accumulation of stable RNAs [15]. Expression of other genes is affected indirectly by the reduced levels of GTP, for example, by favoring transcription of genes whose initiating nucleotide is ATP [19].

Two proteins, RelA and RSH, participate in (p)ppGpp metabolism in *Streptomyces* spp., likely with a division of labor in which RelA and RSH respond to amino acid/glucose starvation and phosphate starvation, respectively [20,21,22]. Starvation of *Streptomyces* spp. results in elevated levels of (p)ppGpp as well as a reduced GTP pool, as also seen in *B. subtilis*, accompanied by reduced transcription of rRNA genes [23,24,25,26]. For example, *S. clavuligerus* in which *relA* is inactivated fails to synthesize (p)ppGpp and maintains steady levels of GTP during growth, whereas wild-type cells exhibit an approximately three-fold decrease in GTP levels as (p)ppGpp accumulates [27]. A similar (p)ppGpp-dependent decrease in cellular GTP levels was previously demonstrated in *S. griseus* during nutritional shift-down, and it was attributed to ppGpp-mediated inhibition of IMP dehydrogenase (GuaB; Figure 1) [28]. Starvation also elicits a complex morphological differentiation in Streptomycetes, a process that involves formation of aerial hyphae and unicellular spores for dispersal, and this differentiation may be accompanied by production of bioactive secondary metabolites. Erection of aerial hyphae requires the *bld* (bald) gene products, while differentiation of aerial hyphae into spores requires the *whi* (white) gene products [29]. The transcriptional regulator BldD controls a large number of genes, many of which are involved in differentiation and antibiotic production [3].

The secondary messenger c-di-GMP, which is synthesized from two molecules of GTP by diguanylate cyclases and degraded by phosphodiesterases, also participates in differentiation. The two *S. coelicolor* proteins RmdA and RmdB possess phosphodiesterase activity and the *rmdA-rmdB* double mutant accumulates three-fold higher levels of c-di-GMP and is completely deficient in aerial hyphae formation [30]. Deficiency in generation of aerial mycelium is also a characteristic of *S. coelicolor* deleted for *cdgB*, which encodes a diguanylate cyclase, and of cells over-expressing *cdgB*; *cdgB* over-expression also inhibits formation of the antibiotic actinorhodin [31]. Expression of *cdgB* is controlled by BldD; notably, DNA binding by BldD is activated by direct binding of c-di-GMP, leading to repression of the BldD-controlled sporulation genes during vegetative growth [32].

## 3. Secondary Metabolism in *Streptomyces*

Members of this genus produce numerous antibiotics or secondary metabolites that are commercially or clinically relevant. These natural products possess, for example, antibacterial, antiviral, anti-tumor, and anti-fungal properties [4]. Secondary metabolites are produced in response to environmental conditions, including starvation or growth of other nearby organisms that may belong to the same or different bacterial genera, or to different species such as plants or fungi; such compounds are therefore thought to furnish the producer with a competitive advantage [34]. Under laboratory conditions, many of the gene clusters that encode proteins involved in generation of natural products are silent. Efforts to elicit their expression by co-culturing *Streptomyces* spp. with various other microbes or by addition of natural and synthetic chemicals have been implemented, and this has resulted in the identification of numerous antibiotics [35]. Combining drug resistant mutations also resulted in activation of antibiotic gene clusters [36,37]. Inducers of secondary metabolism are likely to act indirectly through intracellular signaling, including the generation of alarmones. Since many gene clusters have resisted efforts to elicit their expression during laboratory conditions, *Streptomyces* spp. (and other bacterial species known for production of bioactive secondary metabolites, such as *Burkholderia* spp. [38]) are predicted to constitute a valuable resource for discovery of novel biologically active compounds.

Genes linked to biosynthesis of secondary metabolites generally form very large clusters, often comprising multiple operons, and analysis of genome sequences suggests that the majority are species specific [39]. For instance, *S. coelicolor* strain A3(2) produces several compounds, including the red-pigmented undecylprodigiosin (RED, a member of the prodiginine family of bacterial alkaloids), the deep blue polyketide actinorhodin (ACT), and methylenomycin (Figure 2; reviewed in [40]), while *S. griseus* produces streptomycin, *S. virginia* produces virginiamycin, *S. natalensis* produces pimaricin, and *S. avermitilis* produces avermectin [40]. The antibiotic biosynthetic gene clusters often include regulatory genes referred to as cluster-situated regulators (CSR) [40,41]. The CSRs may directly regulate the transcription of the corresponding antibiotic gene cluster or they may control antibiotic production indirectly through other mechanisms [40]. Activation of antibiotic synthesis is complex and may involve multiple regulators, including factors that affect both antibiotic production and differentiation such as BldD, and it may require a variety of signals, including accumulation of (p)ppGpp and c-di-GMP.

## 4. Link Between Purine Salvage and Production of Guanosine Nucleotide Second Messengers

While (p)ppGpp was first identified under conditions of amino acid starvation, it has since been associated with a number of other stress conditions, most notably conditions faced by bacterial pathogens when they infect a host [42]. Since guanosine nucleotide second messengers are synthesized from GDP or GTP (and at the expense of ATP hydrolysis in case of (p)ppGpp), the enzymes required for purine *de novo* synthesis or salvage are critical for the generation of these second messengers. Direct links between activity of purine metabolic enzymes and (p)ppGpp accumulation have been documented in several bacterial species, including the effects on virulence in the case of pathogens. 

### 4.1. Cellular Levels of GTP and (p)ppGpp are Inter-Dependent

*De novo* purine biosynthesis involves the assembly of the purine ring on phosphoribosyl-pyrophosphate (PRPP) to generate inosine monophosphate (IMP), which is the precursor to both AMP and GMP (Figure 1) [43,44]. Purine bases or nucleotides, including those acquired from exogenous sources, may be inter-converted using salvage pathways. Activity of a number of enzymes in these pathways has been reported to affect the generation of (p)ppGpp, including hypoxanthine guanine phosphoribosyl transferase (HGPRT), which salvages guanine, xanthine, and hypoxanthine nucleobases by converting them to the corresponding mononucleotides, IMP dehydrogenase (GuaB), which catalyzes the committed step in *de novo* GMP biosynthesis, GMP synthetase (GuaA), and xanthine dehydrogenase (Xdh). Xdh converts hypoxanthine to xanthine, thus favoring the conversion of adenine to guanine, and it can divert purines away from salvage pathways and towards degradation by catalyzing the oxidation of xanthine to urate [45]. For example, *guaA* and *guaB* mutants of the Lyme disease pathogen *Borrelia burgdorferi* fail to infect a mouse model, illustrating that purine metabolic enzymes play an important role in virulence [46]. Lacking enzymes involved in *de novo* purine biosynthesis, this bacterium in addition depends on transport proteins to salvage purines from the host [47]. Both (p)ppGpp and c-di-GMP play a vital role during starvation and survival in the tick host, and failure to synthesize these messengers reduces persistence [48,49]. In *Sinorhizhobium meliloti,* (p)ppGpp synthesis is increased upon carbon or nitrogen starvation, concomitant with an increase in the transcript levels of *xdh*. This indicates that the *xdh* gene product is linked to synthesis of (p)ppGpp during starvation [50]. Consistent with this interpretation, *xdh* expression is also increased in stationary phase in *S. coelicolor*, whereas inhibition of Xdh with allopurinol attenuates (p)ppGpp production [51]. In *Listeria monocytogenes*, both *hgprt* and *relA* mutants are incapable of synthesizing (p)ppGpp and fail to cause infection in mice [52]. Evidently, the enzymes of purine metabolism are critical for maintaining the requisite levels of guanosine nucleotide second messengers.

A feedback response in which (p)ppGpp negatively controls activity of purine metabolic enzymes also exists. As noted above, *S. griseus* GuaB is inhibited by ppGpp, which accounts for reduced cellular levels of GTP during stringent response (Figure 3) [28]. In *B. subtilis,* (p)ppGpp inhibits several GTP biosynthesis enzymes, but primarily HGPRT and Gmk, which converts GMP to GDP, in order to regulate the GTP levels in the cell [9]; in absence of (p)ppGpp, excess GTP leads to cell death [18]. *Staphylococcus aureus* HGPRT and Gmk are also direct targets for (p)ppGpp, suggesting a conserved regulatory mechanism [10]. By contrast, Gmk is not specifically inhibited by (p)ppGpp in *Enterococcus faecalis*, whereas HGPRT is [53]. In addition, HPRT/GPRT were recently confirmed as physiologically relevant targets of (p)ppGpp in *E. coli* as well, suggesting that direct regulation of purine metabolic enzymes by (p)ppGpp is not unique to Gram-positive species [8]. That (p)ppGpp targets different enzymes in various bacterial species speaks to distinct mechanisms for control of GTP homeostasis.

### 4.2. Production of Xdh As a Mechanism to Promote Purine salvage

Control of cellular GTP levels is thought to be a primary function of (p)ppGpp in Gram-positives, guarding against excess GTP accumulation during balanced growth and effecting a decrease during the stringent response [9,18,53]. However, excessive depletion of GTP is also deleterious, and mechanisms exist that favor synthesis of guanosine nucleotides. In *S. coelicolor*, xanthine dehydrogenase regulator (XdhR) represses expression of the *xdh* operon; XdhR binds ppGpp (and with lower affinity to GTP), a result of which is attenuated DNA binding and increased expression of the *xdhABC* gene cluster during the stringent response [51]. This mode of regulation would bias purine salvage pathways towards production of guanosine nucleotides, particularly during stringent response when *de novo* pathways are downregulated and the bacteria are dependent upon salvage pathways. The ~0.2 mM affinity of ppGpp for XdhR suggests that *xdh* expression (and hence GTP synthesis) will be induced when (p)ppGpp accumulates to mM concentrations; under these conditions, (p)ppGpp-mediated repression of target enzymes in the purine biosynthesis pathways would be efficient, and production of Xdh may avert excessive depletion of GTP. If (p)ppGpp preferentially inhibits GuaB, as reported in *S. griseus* [28], then GTP production would depend on synthesis of XMP *via* xanthine (Figure 1), rationalizing the need for (p)ppGpp-dependent control of *xdh* expression (Figure 3).

The *xdhR-xdhABC* gene locus is not conserved in *B. subtilis*, where instead expression of *xdh* is induced in the presence of hypoxanthine [54]. By contrast, hypoxanthine is not a ligand for *S. coelicolor* XdhR [51]. This may reflect that *B. subtilis* Xdh has a primary role in purine catabolism, whereas *S. coelicolor* Xdh activity is more important for GTP homeostasis. Consistent with this interpretation, the catabolism of xanthine to urate appears to be disfavored in *Agrobacterium fabrum* during the stringent response when (p)ppGpp accumulates and *xdh* expression is increased; *A. fabrum* encodes a locus that is homologous to the *xdhR-xdhABC* gene cluster found in *Streptomyces* spp. [55]. Furthermore, an *S. coelicolor xdhR* mutant, which overproduces Xdh, is defective in forming aerial mycelium and spores and it overproduces the antibiotic ACT [56,57,58]; the Δ*xdhR* strain does not express *whi* genes, which are important for sporulation, likely on account of increased (p)ppGpp synthesis and dysregulated GTP homeostasis. 

*A. fabrum* XdhR binds c-di-GMP with even higher affinity than ppGpp [51]. As noted above, c-di-GMP is also involved in morphological differentiation and antibiotic production in *Streptomyces* spp., and it activates DNA binding by the key regulator BldD. *S. coelicolor* XdhR likewise responds to c-di-GMP by attenuated DNA binding (Sivapragasam and Grove, unpublished), which may favor synthesis of GTP to sustain production of secondary messengers that derive from GTP. Taken together, we propose that regulation of purine metabolism and GTP homeostasis in Streptomycetes is distinct from that reported in *B. subtilis* and that these processes need to be carefully managed to maintain the optimal synthesis of guanosine nucleotide second messengers. 

Starvation of nutrients, stationary phase, and slow growth leads to degradation of stable RNAs, rRNA and tRNA. These RNAs, which are typically not degraded during balanced growth, account for the vast majority of total RNA, and they represent a valuable source of nutrients [59]. For instance, extensive degradation (>90%) of the 23S rRNA and 16S rRNA occurs in *Salmonella* strains when they reach stationary phase [60]. A similar degradation of rRNA is observed in a *relA* mutant of *E. coli* following nutrient starvation [61]. Even during slow growth, almost 70% of the rRNA that is newly synthesized is degraded. This degradation is considered to be a vital survival strategy as it furnishes the bacteria with a source of nucleotides for salvage pathways during stress or starvation.

## 5. Guanosine Metabolism Impinges on Antibiotic Production

Analysis of annotated *Streptomyces* genomes suggests the presence of >20 species-specific gene clusters implicated in synthesis of secondary metabolites per genome, many of them antibiotics [39]. Production of many such natural products is coordinated with morphological differentiation, however, regulatory mechanisms are complex, and (p)ppGpp has been reported to serve as both negative and positive regulator. 

### 5.1. Production of Actinorhodin (ACT) and Undecylprodigiosin (RED)

Production of antibiotics such as ACT and RED (Figure 2) in *S. coelicolor* generally takes place in stationary phase when (p)ppGpp accumulates (Figure 3) and genes encoding the biosynthetic enzymes are activated by cluster-situated regulators ActII-ORF4 and RedD, respectively [62,63]. Inactivation of *relA* in *S. coelicolor* A2(3) abolishes (p)ppGpp synthesis and production of both ACT and RED under nitrogen-limiting conditions, and this correlates with the reduced accumulation of both *actII*-ORF4 and *redD* transcripts (Table 1) [64]. Under conditions of nutrient sufficiency, induced production of (p)ppGpp results in activation of *actII*-ORF4, but not *redD*, indicating that accumulation of (p)ppGpp *per se* does not induce RED production [21,65]. While inactivation of either *relA* or *rshA* in *S. coelicolor* M600 leads to a complete lack of ACT production in batch culture, RED production is higher than in wild-type cells [20]. Based on the analysis of (p)ppGpp levels and antibiotic production in a strain carrying a mutation in the principal sigma factor (a mutation that was speculated to reduce levels of substrates for (p)ppGpp synthesis), it was inferred that (p)ppGpp-mediated induction of antibiotic production is extremely sensitive to differences in cellular (p)ppGpp pools [66]. In *S. coelicolor* and *S. griseus*, the cyclic AMP (cAMP)-binding protein EshA is essential for ensuring adequate (p)ppGpp synthesis by mechanisms that remain unclear and this, in turn, is required for (p)ppGpp-mediated control of GTP levels. EshA is therefore important for morphological differentiation and ACT/streptomycin production [67]. This analysis also concluded that finely tuned (p)ppGpp accumulation is essential for ensuring antibiotic production.

Transcription of *actII*-ORF4 is under control of numerous regulators, whereas *redD* is activated by RedZ; *redZ* expression, however, is controlled by some of the same factors that regulate *actII*-ORF4 expression (reviewed in [40]). The transcriptional regulator XdhR, which was previously shown to control expression of the *xdh* gene cluster by mediating a (p)ppGpp-dependent increase in *xdh* expression during nutrient limitation and stationary phase [51,56], also binds the promoter regions of *actII*-ORF4 and *actII*-ORF1. The gene *actII*-ORF1 encodes another cluster-situated regulator of ACT biosynthesis. The binding of XdhR leads to the repression of genes encoding ACT biosynthetic enzymes and hence production of ACT in *S. coelicolor* M145 [58]. Since (p)ppGpp is a ligand for XdhR, entry into stationary phase or stringent response would be expected to lead to dissociation of XdhR from the *actII*-ORF4 and *actII*-ORF1 promoters, thus facilitating ACT biosynthesis. Accordingly, XdhR directly links regulation of purine metabolism with (p)ppGpp–mediated control of antibiotic production.

### 5.2. Production of Other Antibiotics

(p)ppGpp has also been implicated in controlling the antibiotic production in other *Streptomyces* spp. For example, an *S. antibioticus relA* mutant is deficient in (p)ppGpp synthesis and actinomycin production, a deficiency that is not alleviated by growth under phosphate-limiting conditions [69]. In *S. griseus*, inhibition of GMP synthetase/GuaA (and hence GTP and (p)ppGpp production) by addition of decoyinine leads to formation of aerial mycelium, but a decrease in the production of streptomycin [23]. A *rel* mutant (tentatively identified as a *relC* mutant) fails to accumulate (p)ppGpp in response to nutrient limitation, and it is deficient in both formation of aerial mycelium and submerged spores and in production of enzymes required for streptomycin biosynthesis. In this *rel* mutant, the addition of decoyinine is accompanied by a marked reduction in cellular GTP levels and it restores the generation of submerged spores; this indicates that morphological differentiation is initiated when GTP levels decrease, but that streptomycin biosynthesis requires (p)ppGpp.

In *S. clavuligerus*, inactivation of *relA* leads to the expected failure to accumulate (p)ppGpp and an inability to sporulate. However, production of clavulanic acid and cephamycin C and transcription of biosynthetic genes is increased in the *rel* mutant, indicating that antibiotic production is negatively regulated by (p)ppGpp [27]. Curiously, the production of these antibiotics in a *relA* mutant strain also appears to depend on strain background and/or growth media as reflected in a separate study in which an *S. clavuligerus relA* disruptant strain was deficient in antibiotic production [71]. The general role of (p)ppGpp in eliciting an upregulation of antibiotic production in a number of bacterial species has also motivated efforts towards metabolic engineering by expressing a (p)ppGpp synthetase. In the actinomycete *Saccharopolyspora erythraea*, this approach yields increased production of erythromycin, but only in a wild-type strain and not in an industrial strain, which already exhibits elevated (p)ppGpp levels [72].

### 5.3. Regulation of Antibiotic Production by c-di-GMP

Both (p)ppGpp and c-di-GMP affect morphological differentiation and regulation of antibiotic production. As noted above, the master regulator BldD represses genes involved in formation of aerial hyphae (the *bld* genes) and sporulation (the *whi* genes) during vegetative growth, and this occurs when BldD binds c-di-GMP [29,32]. Accordingly, c-di-GMP is required for repression of the BldD regulon, and overexpression of the diguanylate cyclases CdgA or CdgB blocks formation of aerial hyphae [3,31]. BldD directly controls expression of *cdgA* and *cdgB*, a regulatory mechanism that may prevent excessive accumulation of c-di-GMP. A *bldD* mutant likewise exhibits a bald phenotype as it bypasses formation of aerial hyphae and sporulates prematurely [32]. The production of antibiotics correlates with these morphological phenotypes as both overexpression of diguanylate cyclases and inactivation of *bldD* leads to a defect in ACT production [73]. BldD controls activity of the *adpA* gene (*bldH* in *S. coelicolor*), which encodes an AraC-family activator of *actII*-ORF4 and *redD* expression, and it binds directly to *nsdA*, which encodes a repressor of ACT production [3,74]. Accordingly, either absence of BldD or hyper-activation of its DNA binding by excess c-di-GMP is likely to result in failure to control expression of cluster-situated activators of ACT and RED. Since XdhR also binds *actII*-ORF4, and since DNA binding by XdhR is attenuated by c-di-GMP (Sivapragasam and Grove, unpublished), we speculate that XdhR may also participate in fine-tuning of *actII*-ORF expression in response to cellular levels of c-di-GMP.

## 6. Conclusions

Antibiotic production in *Streptomyces* spp. is linked to the stringent response and morphological differentiation (Figure 3). During vegetative growth, cellular GTP levels are high, conditions under which synthesis of c-di-GMP would be favored, and BldD–c-di-GMP represses *bld* and *whi* genes. As (p)ppGpp levels increase, the attendant decrease in cellular GTP levels is associated with de-repression of *bld* and *whi* genes and erection of aerial hyphae and sporulation. By contrast, antibiotic production generally occurs as a direct consequence of (p)ppGpp production, with BldD–c-di-GMP playing a poorly understood role in controlling expression of several regulators of antibiotic biosynthetic genes. As both signaling molecules are synthesized from GTP, purine metabolism affects their accumulation, yet GTP homeostasis is likely controlled by distinct mechanisms in different bacterial species, as evidenced for example by the specific purine metabolic enzymes that are direct targets for (p)ppGpp. Notably, recent evidence suggests that the transcription factor XdhR furnishes a direct link between purine salvage and ACT production. Accumulation of (p)ppGpp is associated with increased production of Xdh, as (p)ppGpp attenuates binding of XdhR to the *xdh* promoter; this would circumvent the (p)ppGpp-mediated inhibition of GuaB and allow GTP production to proceed *via* xanthine and XMP. Under these conditions, XdhR-mediated repression of *actII*-ORF4 would also be relieved by (p)ppGpp. Evidently, finely tuned mechanisms operate to control guanosine metabolism, in turn impinging on production of antibiotics.

## Figures and Tables

**Figure 1 antibiotics-08-00076-f001:**
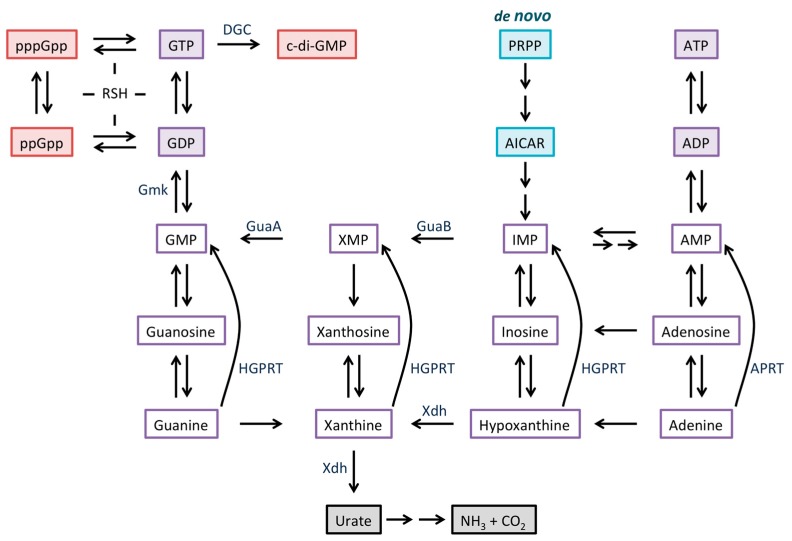
Overview of purine metabolism. The *de novo* biosynthesis (turquoise) starts with phosphoribosyl pyrophosphate (PRPP) and ends with inosine monophosphate (IMP); the intermediate AICAR (5-aminoimidazole-4-carboxamide ribonucleotide) accumulates in presence of dihydrofolate reductase inhibitors [33]. Intermediates in salvage pathways are outlined in purple. In some species, guanine, xanthine, and hypoxanthine are converted to the corresponding mononucleotides by different enzymes. Xdh can initiate degradation of purines (gray). RelA/SpoT Homolog (RSH) family enzymes are responsible for (p)ppGpp synthesis and hydrolysis, while diguanylate cyclase (DGC) synthesizes c-di-GMP (red).

**Figure 2 antibiotics-08-00076-f002:**
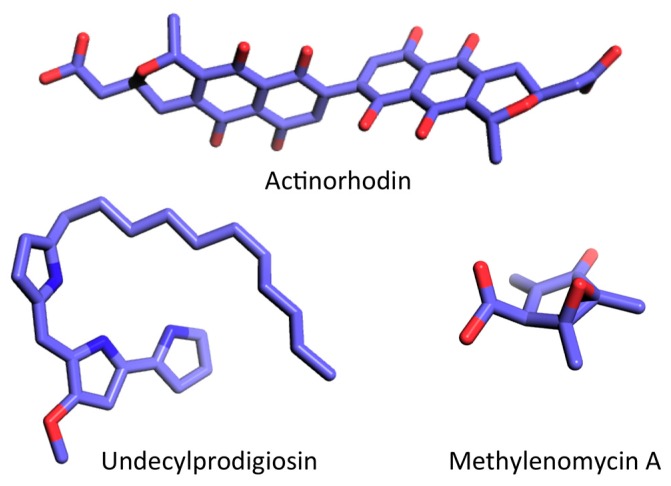
Examples of secondary metabolites produced by *S. coelicolor*. Structures were obtained from the PubChem database and rendered using PyMol (C, light blue; N, dark blue; O, red). Actinorhodin, CID 441143; undecylprodigiosin, CID 135515151; methylenomycin A, CID 122733.

**Figure 3 antibiotics-08-00076-f003:**
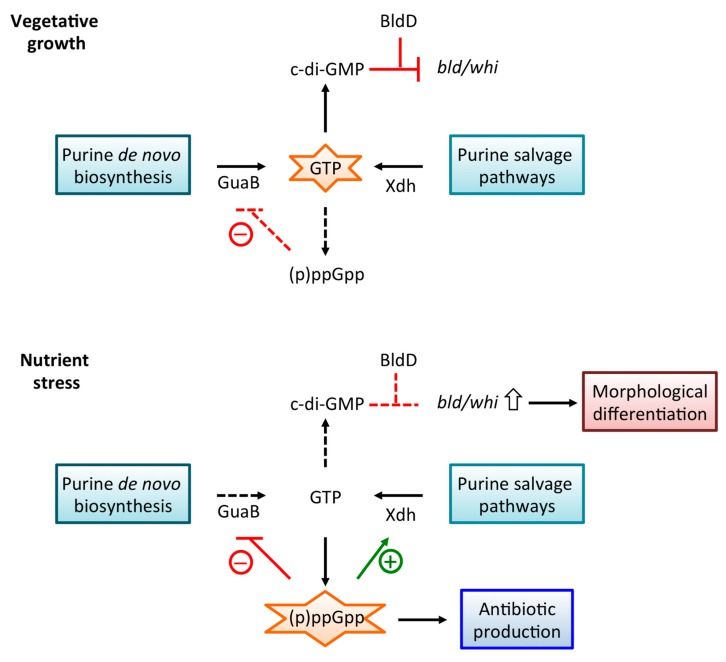
Summary of link between guanosine metabolism and antibiotic production. During vegetative growth, cellular concentrations of GTP are high, but maintained by (p)ppGpp-mediated inhibition of GTP biosynthesis, most likely direct inhibition of GuaB. Under these conditions, c-di-GMP is produced, and it binds BldD to form a complex, which represses *bld* and *whi* genes. Under stress conditions, (p)ppGpp accumulates, resulting in efficient inhibition of GTP biosynthesis. However, high levels of (p)ppGpp also mediate an increase in Xdh production, averting excessive depletion of GTP. Reduced production of c-di-GMP results in derepression of *bld* and *whi* genes, leading to morphological differentiation. The accumulation of (p)ppGpp generally leads to production of secondary metabolites, including antibiotics. The role of BldD in antibiotic production remains to be fully elucidated.

**Table 1 antibiotics-08-00076-t001:** Examples of factors involved in antibiotic production in *Streptomyces* spp. that are linked to stress response or GTP synthesis.

Species	Factor	Link to Guanosine Metabolism or Antibiotic Production
*S. coelicolor*	RelA	(p)ppGpp synthesis, ACT/RED production [64]
*S. coelicolor*	RelC (RplK)	(p)ppGpp synthesis, ACT/RED production [68]
*S. coelicolor*	EshA	(p)ppGpp synthesis, ACT production [67]
*S. coelicolor*	XdhR	Xdh and (p)ppGpp synthesis, ACT production [51,56,58]
*S. coelicolor*	RmdA, RmdB	c-di-GMP degradation, ACT production [30]
*S. coelicolor*	CdgA, CdgB	c-di-GMP synthesis, ACT production [31]
*S. antibioticus*	RelA	(p)ppGpp synthesis, actinomycin production [69]
*S. antibioticus*	RelC	(p)ppGpp synthesis, actinomycin production [70]
*S. clavuligerus*	RelA	(p)ppGpp synthesis, clavulanic acid and cephamycin C production [27,71]
*S. griseus*	Rel GuaA	(p)ppGpp synthesis, streptomycin production [23]

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
