# Peer review of "The Link between Purine Metabolism and Production of Antibiotics in Streptomyces"

_antibiotics, 2019, doi:10.3390/antibiotics8020076_

Round 1

Reviewer 1 Report

This is a prospective and well-written review. In this review, authors have discussed about the mechanisms which control guanosine metabolism and further leads to antibiotic production in Streptomyces spp. There are few corrections which should be addressed:

Please provide references for lines, 293-297.

Please correct grammatical mistakes in manuscript like in line 86.

Author Response

This is a prospective and well-written review. In this review, authors have discussed about the mechanisms which control guanosine metabolism and further leads to antibiotic production in Streptomyces spp. There are few corrections which should be addressed:

Please provide references for lines, 293-297.

Reference 23 was provided (in line 294). 

Please correct grammatical mistakes in manuscript like in line 86.

There is no grammatical mistake in line 86. The “spelling and grammar” search in MS Word did not uncover any grammatical mistakes in the document.

Reviewer 2 Report

This manuscript reviews current knowledge on factors affecting antibiotic production in Streptomyces species, particularly focusing on the purine metabolism effects and the stringent response. Authors discuss mechanisms controlling guanosine metabolism and thus affecting production of antibiotics. 

The manuscript is well written, logically organized with a good flow. 

As a suggestion, I would add a figure with structures of the antibiotics mentioned in Section  3, particularly RED, ACT, and then cited in 5.1 and 5.3. 

Author Response

This manuscript reviews current knowledge on factors affecting antibiotic production in Streptomyces species, particularly focusing on the purine metabolism effects and the stringent response. Authors discuss mechanisms controlling guanosine metabolism and thus affecting production of antibiotics. 

The manuscript is well written, logically organized with a good flow. 

As a suggestion, I would add a figure with structures of the antibiotics mentioned in Section  3, particularly RED, ACT, and then cited in 5.1 and 5.3. 

We have followed the reviewer’s suggestion and included new Figure 2 in section 3 with structures of S. coelicolor-derived antibiotics mentioned. This figure has also been cited in section 5.

Reviewer 3 Report

Current manuscript shows the relation between purine metabolism with antibiotics production through guanosine phosphates biosynthesis pathway. Overall, flow of the review is clear and manuscript prepared well.

However, there are some points require the attention of the authors as below;

1. Current manuscript present only one figure about the purine metabolism, which does not cover the main idea of the manuscript. Suggesting more figure(s) which covers whole flow of the manuscript and other detailed between guanosine metabolism and antibiotic production are needed.

2. Some of the part such as [stringent respons in Gram negative] seams not proper for the manuscript, which covers too wide. Some efforts to organize this is needed.

Author Response

Current manuscript shows the relation between purine metabolism with antibiotics production through guanosine phosphates biosynthesis pathway. Overall, flow of the review is clear and manuscript prepared well.

However, there are some points require the attention of the authors as below;

1. Current manuscript present only one figure about the purine metabolism, which does not cover the main idea of the manuscript. Suggesting more figure(s) which covers whole flow of the manuscript and other detailed between guanosine metabolism and antibiotic production are needed.

As suggested by Reviewer 2, we have added a figure with structures of the S. coelicolor-derived antibiotics mentioned in the text. We have also included new Figure 3, which summarizes key links between guanosine metabolism and antibiotic production.

2. Some of the part such as [stringent respons in Gram negative] seams not proper for the manuscript, which covers too wide. Some efforts to organize this is needed.

The consideration of stringent response in Gram negatives is limited to one paragraph. We suggest that it is relevant as it alerts the reader to the differences between Gram-negative and Gram-positive species. No change in revision.